# Influence of Additional White, Red and Far-Red Light on Growth, Secondary Metabolites and Expression of Hormone Signaling Genes in Scots Pine under Sunlight

**DOI:** 10.3390/cells13020194

**Published:** 2024-01-19

**Authors:** Pavel Pashkovskiy, Mikhail Vereshchagin, Alexander Kartashov, Yury Ivanov, Alexandra Ivanova, Ilya Zlobin, Anna Abramova, Darya Ashikhmina, Galina Glushko, Vladimir D. Kreslavski, Vladimir V. Kuznetsov

**Affiliations:** 1K.A. Timiryazev Institute of Plant Physiology, Russian Academy of Sciences, Moscow 127276, Russia; pashkovskiy.pavel@gmail.com (P.P.); mhlvrh@mail.ru (M.V.); botanius@yandex.ru (A.K.); ivanovinfo@mail.ru (Y.I.); aicheremisina@mail.ru (A.I.); ilya.zlobin.90@mail.ru (I.Z.); ann.kiedis2000@gmail.com (A.A.); ashikhminada@yandex.ru (D.A.); g.glushko-v@yandex.ru (G.G.); 2Institute of Basic Biological Problems, Russian Academy of Sciences, Pushchino 142290, Russia; vkreslav@rambler.ru; 3Department of Plant Physiology, Biotechnology and Bioinformatics, Biological Institute, National Research Tomsk State University, Tomsk 634050, Russia

**Keywords:** Scots pine, red light, far-red light, terpenoids, gene expression, growth parameters

## Abstract

The influence of short-term additional white (WL), red (RL) and far-red (FRL) light and combined RL+FRL on the physiological morphological and molecular characteristics of two-year-old Scots pine plants grown in a greenhouse under sunlight was studied. Additional RL and RL+FRL increased the number of xylem cells, transpiration and the expression of a group of genes responsible for the biosynthesis and signaling of auxins (*AUX/IAA*, *ARF3/4*, and *ARF16*) and brassinosteroids (*BR-α-RED* and *BRZ2*), while the expression of genes related to the signaling pathway related to jasmonic acid was reduced. Additionally, WL, RL and RL+FRL increased the content of proanthocyanidins and catechins in young needles; however, an increase in the expression of the chalcone synthase gene (*CHS*) was found under RL, especially under RL+FRL, which possibly indicates a greater influence of light intensity than observed in the spectrum. Additional WL increased photosynthetic activity, presumably by increasing the proportion and intensity of blue light; at the same time, the highest transpiration index was found under RL. The results obtained indicate that the combined effect of additional RL+FRL can accelerate the development of pine plants by increasing the number of xylem cells and increasing the number of aboveground parts but not the photosynthetic activity or the accumulation of secondary metabolites.

## 1. Introduction

Plants can perceive light and respond to it through changes in metabolism and morphogenesis. One of the key elements of a plant’s response to light is its response to the ratio of red light (RL) to far-red (FRL) light (RL/FRL). This ratio is inextricably linked to hormonal regulation and other components of the multifaceted network of cross-signals that control the growth, development and adaptation of plants, particularly forest plants [1]. Forests are complex ecosystems, where light is one of the main factors influencing the growth and development of forest plants. In forest phytocenoses, the RL/FRL ratio in the shade of trees notably decreases since the red component of the spectrum of sunlight is largely absorbed by the pigments of the leaves of the upper layer of trees [2], while a significant portion of green (500–550 nm) and far-red (>700 nm) light is absorbed by the leaves of plants of the lower and middle layers [3]. As a result, the RL/FRL ratio in dense forests decreases compared to that in open areas, where the RL/FRL ratio at midday is close to 1 [4]. A change in the RL/FRL ratio becomes a signal for most plants in the lower layers, affecting their physiological functions (shade avoidance syndrome, SAS). Moreover, the young seedlings of trees in the upper layer also experience FRL effects because they are vegetated under the canopy of mother plants [5].

Phytochromes are chromoproteins responsible for the perception of RL and FRL. A higher RL/FRL ratio indicates that a larger fraction of phytochrome molecules are active (Pfr), initiating downstream signaling that regulates various physiological processes. Thus, phytochromes regulate seed germination, stem elongation, leaf morphology formation and the transition to generative development [1]. In this case, phytochrome in its active form interacts with the signaling molecules that are involved in light signaling, the corresponding components of which, such as transcription factors, in turn, influence the expression of genes associated with these proteins, regulating the biosynthesis of various proteins, particularly those involved in hormonal regulation [6].

Depending on the intensity and spectral composition of the light reaching the vegetation of the forest floor, plants use different adaptation strategies [7]. Light-loving plant species, which include Scots pine, are adapted to life under conditions of high RL/FRL ratios. However, in the first years of life, these plants are able to grow even with a lack of RL under the forest canopy. When plants grow in sunlight at a ratio of RL to FRL close to 1, the spectrum of incident radiation may contain a relatively high proportion of the blue component of light, which acts on plants both directly and indirectly through photoreceptors that absorb UV-A and BL cryptochromes [8].

In addition to photoreceptors and light signaling components, hormones play an important role in the regulation of physiological functions by stimulating plant growth, thereby providing competitive advantages for a particular individual during a “struggle” for light in forest phytocenosis [9]. A low RL/FRL ratio in most heliophyte plants induces the formation of SAS [10], which manifests in the elongation of stems and petioles, a reduction in leaf area and early flowering. Moreover, auxins and gibberellins (GAs) accumulate, which promotes cell elongation [10]. In this case, auxins are actively involved in the response of plants to changes in the RL/FRL ratio. High levels of RL/FRL support auxin synthesis and polar auxin transport, promoting shoot elongation and apical dominance. A low RL/FRL ratio, which signals shading, triggers a shadow avoidance response, which is accompanied by the activation of auxin signaling [10]. Cytokinins often counteract the effects initiated by auxins. When the RL/FRL ratio is high, cytokinins stimulate cell division and differentiation, especially in roots, thereby promoting balanced growth [11]. Brassinosteroids (BRs) enhance auxin action at low RL/FRL ratios, promoting cell elongation and division and helping to overcome shading [12]. Jasmonates (JAs) are involved in plant defense reactions against biopathogens, but their content and activity can also be affected by the RL/FRL ratio. Low RL/FRL values, which cause SAS, can reduce resistance to pests; however, increased levels of JA prepare plants for such potential threats [13]. The subtleties of light and hormone signaling pathways lie first in their interaction. The synthesis and degradation of hormones under the influence of light may additionally influence components of light signaling. For example, the levels of auxins, which are affected by the RL/FRL ratio, can alter the stability of phytochrome (PHY) signaling components [14]. In addition, transcription factors (TFs), such as phytochrome-interacting factors (PIFs), play an important role in the integration of light and hormonal signals. For example, PIFs stabilized at low RL/FRL promote the synthesis of auxins and GA.

Light and hormonal signaling induce changes in plant secondary metabolism. Secondary metabolites, which usually include terpenoids, alkaloids and phenolic compounds, play a decisive role in the adaptation of plants to the environment [1]. Changes in light quality, particularly in the RL/FRL ratio and blue light (BL)/RL, may serve as a signal to pine plants to adjust their physiological processes and secondary metabolite production to better suit current conditions. For example, changes in the RL/FRL ratio can affect the formation of phenolics and other compounds, such as terpenoids [15]. Terpenoids, a large and diverse class of secondary metabolites, are known for their role in plant responses to stress and changes in light quality [16]. Terpenoids, particularly carotenoids, in pine plants are largely associated with a protective function; they play a protective role in the presence of excess light, which causes photodamage [17]. Thus, when pine plants are exposed to sudden intense light, the biosynthesis of specific terpenoids may increase since they have antioxidant activity and reduce the level of reactive oxygen species [1]. However, the plant may perceive shaded conditions as a potential threat (for example, from competing plants). In response, the synthesis of specific terpenoids, such as pinene and limonene, which are involved in plant protection from pests, including fungi, may increase [18].

There is no doubt that hormones and secondary metabolites play important roles in ensuring light reactions and increasing plant resistance; however, the key role in the regulation of physiological and morphological reactions in plants is the interaction of light and hormonal signaling pathways. Signaling pathways such as the JA and BR pathways can act together to synergistically or antagonistically regulate plant growth and defense in response to changing environmental factors [19]. In addition, although BR is known for its ability to promote growth, it can also interact with JA, thereby providing a balance between growth and protective functions, which ensures the optimal use of available energetic and structural resources [20]. The evolutionary advantage provided by these hormonal interactions and secondary metabolites becomes especially important in the face of changing habitats. Since many plants have evolved from open to understory environments, quickly adapting to different RL/FRL ratios is essential [21]. This requires the formation of appropriate photoreceptor and hormonal systems that could provide flexible and adequate responses to changes in the environment. The forest canopy, while serving some protective function, also poses problems due to changes in light quality, especially in the RL/FRL ratio, which a plant may encounter several times during its ontogeny. The subtle interaction between the active form of phytochrome—regulated by the RL/FRL ratio—components of the hormonal system and secondary metabolites emphasizes the adaptability and resistance of plants to various light conditions. By controlling the quality of light in greenhouses and on vertical farms, it is possible to change the phenotypic traits, flowering times and growth parameters of plants, optimizing productivity and the use of available resources. All of these factors can form the basis for the creation of modern agriculture, tailored to the specific needs of specific crops.

In previous works, we found that narrowband RL had a specific effect on Scots pine seedlings and plantlets [1,2,22]. We hypothesized that the RL activation of various PHY and auxin signaling components stimulates an increase in the number of xylem cells [2]. RL can regulate stem growth via the effects of phytohormones and secondary metabolites [1,22]. However, the influence of light quality on physiological responses and associated gene expression in Scots pine requires further study. It is currently not known whether such processes manifest in adult plants when FRL is exceeded in the spectrum or when the RL/FRL ratio is different. Thus, FRL can change the ratio of active auxins, affecting cell elongation [23].

In this study, we aimed to determine how additional light with a certain RL/FRL ratio compared to that under normal conditions can affect the size and number of xylem cells in Scots pine, as well as other growth parameters and photosynthetic processes, and whether it is possible to additionally activate the spring growth of shoots with the help of this supplement. We hypothesized that not only auxins, cytokinins and GAs but also phytohormones of the steroid class, such as BR, are involved in the response of plants to additional light. In addition, the possible impacts of additional RL, RL+FRL and WL on the levels of secondary metabolites, such as phenols and terpenoids, as well as the expression of the main genes involved in hormonal signaling and the biosynthesis of the main secondary metabolites, were considered.

## 2. Materials and Methods

### 2.1. Plant Material and Experimental Design

The experiment involving additional irradiation was carried out under the greenhouse conditions of the K.A. Timiryazev Institute of Plant Physiology RAS Moscow (GPS 55.841167, 37.586320). The experiment was carried out on 2-year-old saplings of Scots pine (*Pinus sylvestris* L.), which were dormant after the winter period and growing in 0.7 L pots filled with soil substrate (peat, sand, ground limestone, 350 mg/L N, 30 mg/L P, 400 mg/L K, pH 5.5–6.0). The experiment was carried out from 1 April 2022 to 10 May 2022. The duration of daylight hours was 13–16 h (on average, sunrise at 5:00 am and sunset at 20:00). The sun was at its zenith on average at 12:30, and the average intensity of sunlight during this period was 250 ± 100 μmol m^–2^s^–1^. A shade net was used to diffuse direct sunlight. Additional light was applied for 14 h from 7:00 to 21:00, and the intensity of the light flux under any additional illumination was 250 ± 20 µmol m^–2^s^–1^. The plants were illuminated with diode matrices (Epistar, Taiwan), with red light (RL) at 660 nm (250 µmol m^–2^s^–1^), far-red light (FRL) at 720 nm (250 µmol m^–2^s^–1^), broadband white LEDs (WL) (250 µmol m^–2^s^–1^) and red + far-red light (RL+FRL) at 660 nm + 720 nm (125 + 125 µmol m^–2^s^–1^). The RL/FRL ratio under sunlight and white LED lamps was 1 for RL-1.7, 0.4 for FRL and 0.5 for RL +FRL (Figure 1). The average temperature in the greenhouse was 24 ± 5 °C during the day and 17 ± 5 °C during the night, and the average humidity was 50–80%. The plants were watered regularly.

### 2.2. Morphological Analyses

At the end of the experiment, the following plant growth characteristics were measured: the aboveground height, current-year central shoot height, current-year central shoot height contribution to the aboveground height, average length of current-year lateral shoots, number of current-year lateral shoots, fresh weight of the aboveground part, increase in the fresh weight of the aboveground part, increase in the dry weight of the aboveground part, contribution of the current-year shoot weight to the total weight of the aboveground part, primary root length, fresh weight of the current-year roots and contribution of the current-year root weight to the total root weight and water content of the current-year lateral shoots.

### 2.3. Anatomical Studies

Two-centimeter stem fragments from the central parts of the current-year and previous-year shoots were fixed in 3% paraformaldehyde (in PBS, pH 7.0.) and were used to prepare a series of xylem cross-sections. Cross-sections (0.1 mm thick) were made using a manual microtome and a razor blade. The cross-sections were stained with 0.05% toluidine blue in 0.1 M phosphate buffer at pH 7.0. [24]. The sections were photographed under a light microscope (MicMed-6; LOMO, St. Petersburg, Russia) on a 12 Mpx digital camera (Sony, Tokyo, Japan) at 20× magnification. The obtained images were analyzed with ToupView software Ver 3.7.

The following anatomical parameters were determined for both previous-year shoots and current-year shoots: the average stem diameter, cross-sectional area of the stem, cross-sectional area of the bark, cross-sectional area of the current-year and previous-year xylems, number of xylem cells in the radial row of the current-year xylem and average outer diameter of xylem cells in the current-year xylem.

### 2.4. Definition of Transpiration and Fluorescence Chlorophyll

Midday stomatal conductance to water vapor (*g_sw_*) was measured on previous-year needles using an LI–600 flow-through differential porometer (LI-COR, Inc., Lincoln, NE, USA) between 11:00 and 13:00 h. The surface area of the needles used for g*_sw_*measurements was calculated, and gas exchange values were recomputed with the exact needle area. For details, see Zlobin et al., 2022 [25].

Fluorescence induction curves were measured both on previous-year and current-year needles using a mini-PAM fluorometer II (Walz, Effeltrich, Germany) in dark-adapted plants, as previously described [26]. After a pulse of saturating light, the plant leaves were allowed to adapt to 30 min of darkness, kept in the dark for one minute and then exposed to actinic light for 5 min, followed by pulses of saturating light. Blue LEDs (maximum wavelength of 450 nm) produced the measuring light (0.5 μmol photons m^–2^s^–1^), actinic light (250 μmol (photons) m^–2^s^–1^, duration of 10 min) and saturating pulses (maximum 450 nm, 3000 μmol photons m^–2^s^–1^, 800 ms duration). Parameters based on fluorescence data were determined using Imaging Win v.2.41a software (Walz, Effeltrich, Germany). The values of F_0_, F_v_, F_m_, F_m_’ and F_0_′ were determined. Here, Fm and Fm’ are the maximum levels of chlorophyll fluorescence under dark- and light-adapted conditions, respectively. F_v_ is the photoinduced change in fluorescence, and F_t_ is the level of fluorescence before the saturation pulse was applied. F_0_ is the initial level of chlorophyll fluorescence. On the basis of these results, the maximum (F_v_/F_m_) and effective Y(II) (F_m_’ − F_t_)/F_m_’ PSII photochemical quantum yields and non-photochemical quenching (NPQ) (F_m_/F_m_’ − 1) were determined. We also determined the values of the Y(NO) and Y(NPQ) quantum yields of non-regulated and regulated non-photochemical energy dissipation in PSII, respectively, using PAM–fluorometer software WinControl-3.

### 2.5. Phenolic Compounds

Phenolic compounds were extracted with 80% methanol from samples of roots or needles ground in liquid nitrogen.

The low-molecular-weight antioxidant capacity of Trolox equivalent antioxidant capacity (TEAC) was determined spectrophotometrically according to the method described by Re et al. (1999), involving the reaction of methanolic extracts with 2,2′-azino-bis [3-ethylbenzothiazoline-6-sulfonic acid] diammonium salt (ABTS) (Sigma, Burlington, MA, USA, CAS number 30931-67-0) [27].

The content of total phenolic compounds was determined spectrophotometrically using Folin and Ciocalteu’s phenol reagent (Sigma-Aldrich; MDL number MFCD00132625) according to the procedure described by Singleton and Rossi 1965 [28]. The total phenolic content is expressed as gallic acid equivalents (GAE) in milligrams per gram of fresh weight (FW).

The total flavonoid content was measured according to the methods of Kim et al., 2003 [29]. Afterwards, 1000 µL of distilled water, 150 µL of extracted sample and 50 µL of 5% NaNO_2_ were mixed together. After 6 min, 50 µL of 10% AlCl_3_ was added, and after another 5 min, 300 µL of 1 M NaOH was added to the mixture. The reaction mixture was homogenized, and after 10 min, the absorbance at 510 nm was measured. The total flavonoid content was calculated by constructing a calibration curve using (+)-catechin hydrate (Sigma, CAS Number 225937-10-0) and is expressed as milligrams of (+)-catechin per gram of FW.

The total content of catechins and proanthocyanidins (PAs) was determined spectrophotometrically by the reaction of catechins, PAs and 1% vanillin in acidic media [30]. The catechin and PA contents were calculated by constructing a calibration curve using (+)-catechin hydrate and are expressed as milligrams of (+)-catechin per gram of FW.

Additionally, the PA content was determined via reaction with a butanol reagent. The butanol reagent was prepared by mixing 128 mg of FeSO_4_·7H_2_O and 5 mL of HCl together, and the reaction was completed in 100 mL of n–butanol. Fifty microliters of extracted sample and 700 μL of butanol reagent were mixed together, after which the mixture was heated at 95 °C for 45 min [31]. The sample was cooled, after which the absorbance at 550 nm was measured. The total PA content was calculated by constructing a calibration curve using cyanidin chloride (PhyProof^®^, PHL80022) and is expressed as cyanidin equivalents in milligrams per gram of FW.

### 2.6. Analysis of Total Terpenoids

The frozen needles were ground in a mortar in the presence of liquid nitrogen, and a sample of 100–150 mg was taken. The samples were freeze-dried in a Gyrozen HyperCOOL HC3110 freeze dryer for 24 h. The dried samples were weighed and extracted with 1.5 mL of hexane:ethanol (7:3), initially on a rotary shaker for 10 min and then in an ultrasonic bath for 10 min. The supernatant was separated via centrifugation at 7000× *g* for 10 min. The precipitate was extracted again in the same way. The total supernatant was evaporated to dryness in a vacuum concentrator and dissolved in 1.5 mL of methanol in the dark for 12 h. A terpenoid analysis was performed according to the methods of [32], with modifications. Then, 0.7 mL of the extract and 0.3 mL of distilled water were added to the test tubes, which were subsequently cooled on ice. Then, 0.5 mL of a solution of 1.6% vanillin in concentrated sulfuric acid was added dropwise, and the mixture was cooled with ice. Then, the tubes containing the reaction mixture were cooled, mixed by vortexing and incubated at 60 °C for 20 min. The absorbance of the cooled samples was measured with a Genesys 10S UV-Vis spectrophotometer (Thermo Fisher Scientific, Waltham, MA, USA) at a wavelength of 608 nm. The value of the blank sample, which contained methanol instead of the sample, was also measured. The total terpene and terpenoid contents were calculated from a calibration curve constructed using solutions of α–pinene in methanol [32].

### 2.7. RNA Extraction and RT-PCR

RNA isolation was performed according to the method of [33] modified by [34]. The quantity and quality of the total RNA were determined using a NanoDrop 2000 spectrophotometer (Thermo Fisher Scientific, USA). cDNA synthesis was performed using an M–MLV Reverse Transcriptase Kit (Fermentas, CA, USA) and the oligo (dT) 21 primer. The expression patterns of the genes were assessed using a CFX96 Touch™ Real–Time PCR Detection System (Bio-Rad, Hercules, CA, USA). The gene-specific primers used were as follows (Appendix A): auxin response factor 16 (*ARF16*, KY914544.1), auxin-induced protein 1 *(AUX/IAA*, AY289600.1), jasmonate–Zim domain 1 (*JAZa*, EF083399.1) and histidine-containing phosphotransfer 1 (*HPT1*, ALN42232). Steroid 5-α-reductase DET2 (*BR–α–RED,* A9NWW4), the brassinosteroid-mediated signaling pathway gene (*BRZ2*, MH017214.1), monoterpene synthases such as TPS–mono1 (*mono-TERP*, JQ240296.1), α-terpineol synthase TPS–α-terp (*α-TERP*, JQ240308.1), chalcone synthase (*CHS*, MA_10426264g0020), auxin response factor 3/4 (*ARF3/4,* FN433184.1), *Actin1* (*ACT1*, CBB44933.1) and glyceraldehyde phosphate dehydrogenase (*GAPDH* gene, L26923.1) were selected using nucleotide sequences from the National Center for Biotechnology Information (NCBI) database (www.ncbi.nlm.nih.gov, Bethesda, MD, USA) with Vector NTI Suite 9 software (Invitrogen, Carlsbad CA, USA). The gene transcript levels in the needles and cambium were normalized to the expression of the *Actin1* gene. The gene transcript levels in the roots were normalized to the expression of the *GAPDH* gene. The experiments were performed with 6 biological and 3 analytical replicates. The relative gene expression signal intensity in the WFL plants was considered to be 1.

### 2.8. Statistics

Each plant was treated as an independent biological replicate. For morphological and anatomical studies, 21 plants were analyzed per experimental variant. Each plant sample fixed in liquid nitrogen was treated as a biological replicate; therefore, there were 6 biological replicates for pigment, phenolic compound, terpenoid content and gene expression analyses.

The data were statistically analyzed using SigmaPlot 12.3 (Systat Software, Palo Alto, CA, USA) with a one-way analysis of variance (ANOVA), followed by Duncan’s multiple range post hoc test for normally distributed data (in the figures, significant differences are denoted by different normal letters), and Kruskal-Wallis one-way ANOVA on ranks, followed by the Student-Newman-Keuls post hoc test for nonnormally distributed data and data with unequal variance (in the figures, significant differences are denoted by different italic letters for the Student–Newman-Keuls post hoc test). Different letters indicate significance at *p* < 0.05. The values presented in the tables and figures are the arithmetic means ± standard errors.

## 3. Results

### 3.1. Morphometric Indicators

The quality of the light used in the experiments had a significant impact on a set of morphometric parameters. Thus, the maximum height of the plants was observed when they were irradiated with FRL or RL+FRL light (Figure 2).

Moreover, the current-year central shoot height, the contribution of the current-year central shoot height to the aboveground part height and the water content in the current-year lateral shoots were the greatest when the plants were illuminated with FRL (Figure 2). The number of current-year lateral shoots and the dry weight of the aboveground parts increased, and, as a consequence, the contribution of the current-year shoot weight to the total weight of the aboveground parts was the greatest in the RL and RL+FRL treatments (Figure 2). When illuminated only with RL, the fresh weight of the current-year roots increased; however, other indicators associated with the growth of the root system did not differ from those associated with the other light regimes. Unlike other light regimes, simultaneous RL+FRL was accompanied by an increase in the height of the aboveground parts and an increase in the fresh weight of the aboveground parts of the plants (Figure 2 and Figure 3).

### 3.2. Anatomical Studies

Anatomical analyses of the previous-year shoot showed the maximum stimulating effect of RL on several anatomical indicators, such as the ratio of the current-year xylem area to the entire xylem area (minus the pith), the average number of xylem cells in the radial row of the current-year xylem and the cross-sectional area of the current-year xylem (Figure 4). FRL caused an increase in the contribution of the bark area to the cross-sectional area of the stem, and RL and RL+FRL caused an increase in the cross-sectional area of the stem (Figure 4). Moreover, RL+FRL led to an increase in the diameter of the shoot in the previous year (Figure 4).

The results of the anatomical analyses of the current-year shoot that formed during the experiment were less pronounced because, in the RL variant, only an increase in the average diameter of one xylem cell of the current-year shoot was observed (Figure 4B). The main changes occurred under the combined treatment (RL+FRL), which caused an increase in the ratio of the cross-sectional area of the current-year and previous-year xylems, the average number of xylem cells in the radial row of the current-year xylem and the average length of the radial row of the current-year xylem (Figure 4B). The remaining parameters did not differ significantly among the light regimes.

### 3.3. Total Phenolic Compounds, Low-Molecular-Weight Antioxidants, Flavonoids, Catechins, Proanthocyanidins, Terpenoids and Pigments Were Identified

During the experiments, a complex of biochemical parameters was analyzed in the previous-year needles, as well as in the current-year needles and young (current-year) roots (the sum of phenolic compounds and low-molecular-weight antioxidant capacity (Trolox equivalent antioxidant capacity (TEAC)), flavonoids, catechins, proanthocyanidins and terpenoids). Notably, no significant differences were found in either the roots or the previous-year needles (Table 1, Table 2 and Appendix A). Moreover, an increase in catechins and proanthocyanidins was observed in the current-year needles in all the treatment groups, with the exception of the control and FRL groups (Table 1, Table 2 and Appendix A). Notably, there was a tendency toward an increase in most of the studied parameters in the RL variant, and the most significant changes were observed in the TEAC values of the current-year roots and needles (Table 1 and Table 2).

In the RL treatment, a decrease in all photosynthetic pigments was observed in the current-year needles relative to those in the control and other light options (Table 2). The level of pigments in the previous-year needles did not differ among the treatments (Appendix A). The quality of the supplemental lighting had a noticeable impact on the chlorophyll ratio (Chl *a*/Chl *b*) and carotenoid-to-chlorophyll ratio (Car/Chl *a* + *b*). Specifically, in the current-year needles, the Chl *a*/Chl *b* ratio was the highest in the RL treatment and the lowest in the FRL treatment (Table 2). The variant RL+FRL exhibited an intermediate value, positioned between the effects of RL and FRL. In contrast, in the previous-year needles, the Chl *a*/Chl *b* ratio remained unchanged across the different lighting conditions (Appendix A). The Car/Chl *a* + *b* ratio was the highest in the current-year needles when under RL. However, for the previous-year needles, there were no significant differences in this ratio among the various lighting conditions, except for a noticeable decrease under WL light (Appendix A).

### 3.4. Fluorescent Parameters and Stomatal Conductance

PSII activity in the current-year needles, estimated as the effective quantum photochemical yield of PSII–Y(II), was the highest with additional WL and the lowest with additional FRL. Moreover, the opposite trend was observed for the Y(NO) values. It was the lowest under WL and the highest under FRL. The values of other parameters, such as NPQ and F_v_/F_m_, differed little from each other (Figure 5A,B and Appendix A).

In contrast to that in the current-year needles, the PSII activity in the previous-year needles was the highest under FRL and in the control and minimal with additional WL. The value of F_v_/F_m_ was also the highest in the RL+FRL variant in the previous-year shoots (Appendix A). The values of other parameters, such as NPQ, Y(NPQ) and Y(NO), differed little from each other (Appendix A).

The level of stomatal conductance was the highest in the RL variant, being 2 times greater than that in the other variants (Figure 5B).

### 3.5. Gene Expression

Gene expression was studied in the current-year needles, as well as in the cambium of the previous-year shoots and current-year roots. Notably, RL and the joint action of RL+FRL had stimulating effects on most of the studied genes, especially on the needles (Figure 6A–C and Appendix A).

Thus, in the current-year needles under additional light conditions (RL+FRL), the expression of the genes for the auxin signaling agent *ARF16*, *AUX/IAA*, biosynthesis of jasmonic acid *JAZa* (negative regulation), and biosynthesis and signaling of the brassinosteroids *BR-α-RED* and *BRZ2* increased significantly relative to those in the control, WL and FRL treatments. Moreover, under RL conditions, the expression of these genes was also high but was significantly lower than that in the RL+FRL variant (Figure 6A).

In the features of the cambium, there were no changes in the expression of α-terpineol synthase (*α-TERP*) or monoterpene synthase (*mono-TERP*) and a reduced expression of the *CHS* gene in the RL+FRL variant (Figure 6B). Moreover, in the RL+FRL variant, an increase in the expression of the *ARF3/4* gene was observed, while the expression of this gene was also detected in the roots (Figure 6A–C).

In the roots, the transcript levels of the analyzed genes also increased under conditions of combined exposure to RL+FRL, whereas under supplemental conditions involving only FRL, the expression of these genes was slightly lower (Figure 6C). A distinctive feature of the root response to additional light was an increase in the expression of the cytokinin signaling gene *HPT1*, as well as the absence of changes in the expression of the jasmonate biosynthesis gene *JAZa*. It should also be noted that the level of expression of the auxin signaling gene *ARF3/4* in the current-year roots was the highest in the FRL and RL+FRL variants, in contrast to the findings in other studied organs and tissues (Figure 6C).

## 4. Discussion

The effects of RL and FRL have been extensively studied in angiosperms at the physiological, biochemical and molecular levels [35]. These effects have been studied to a lesser extent in coniferous plants. In the case of Scots pine, when exposed to physiologically active FRL at 720 nm, seedlings exhibit elongation similar to the effect of etiolation [1,36,37]. Our studies also indicate that FRL light stimulates the elongation of central shoots (current-year central shoot height; Figure 2), probably due to cell elongation and an increase in the percentage of water content (77.2 ± 0.3) (Figure 2 and Figure 3). The increase in growth through cell extension under supplemental FRL was substantiated by the observed increase in parameters such as the aboveground plant height, current-year central shoot height and contribution of the current-year central shoot height to the aboveground part height under these light conditions. This difference was additionally confirmed by the relative decrease in fresh biomass compared to in the RL treatment (Figure 2 and Figure 3). This finding is consistent with the high FRL expression of genes associated with the biosynthesis of auxin inducers involved in cell elongation and signaling, as well as the increased percentage of water in the current-year lateral shoots (Figure 2 and Figure 6A).

We previously demonstrated that RL and FRL have effects on the morphological physiological responses of Scots pine plants similar to those on herbaceous plants [1]. It is important to note that the responses of plants grown in a climate chamber and exposed to a specific spectrum of light from the moment of germination may differ from the reactions of adult plants transferred to an environment with a varied spectrum of light. It can be assumed that the previous-year shoot of a plant formed under natural light and temperature conditions remembers this phenomenon at the molecular level. Moreover, winter dormancy can lead to the elimination of photoreceptor signals, and the greatest effect can be observed on young needles [38]. Considering that the set of PHYs of conifers differs from that of flowering plants, it can be expected that the reception of RL and FRL is carried out by different types of PHYs, in contrast to some liver mosses, where there is one type of PHY and whose functions have not yet been elucidated [1]. Additional RL and FRL lead to changes in the ratio of active to inactive PHYs, which partially explains the reactions that we observed. One of these reactions is an increase in plant height under RL+FRL and FRL compared to under WL. The contribution of the current-year central shoot height to the aboveground part height and the contribution of the current-year shoot weight to the total weight of the aboveground part, however, only increased for the RL+FRL option; for the FRL option, these indicators were not different from those of the WL option (Figure 2 and Figure 3). In our experiments, RL was shown to increase the cross-sectional area of the stem beginning in the previous year (Figure 4). Notably, this increase was primarily attributed to the enlargement of the xylem area under RL exposure (Figure 4). Concurrently, within the current-year shoot exposed to RL, the xylem area was among the smallest (Figure 4B). However, the effect of the combination of RL+FRL surpassed that of both RL and WL in terms of xylem area. Nonetheless, there were no significant differences between the other lighting conditions and the control group (Figure 4). There was an increase in the number of xylem cells in the current-year shoot under the influence of RL+FRL, which indicates the activation of cell division, while RL reduced the number of cells in the current-year shoot and simultaneously increased their diameter (Figure 4B). In other words, the response of the young (current-year) shoots of Scots pine to the effects of RL and FRL is similar to the response of dicotyledonous plants. Moreover, in the previous-year shoots, RL caused an increase in the cross-sectional area of the xylem but not in that of the bark (Figure 4). Notably, the effects of combined exposure to RL+FRL on the stem from the previous year were less pronounced; however, for the current-year stem, the most significant changes were observed for the variant exposed to both red light and far-red light (RL+FRL) (Figure 4, Table 1). This difference likely indicates that the lower lignified part of the shoot responded to a greater extent to RL than to FRL since combined additional illumination with RL+FRL removed the stimulating effect of RL. This conclusion is in agreement with our previous results obtained for Scots pine seedlings grown in a hydroculture [2].

Variations in the proportions of RL and FRL affect not only the morphological parameters of plants but also the photosynthetic activity associated with photosystem II. It was previously shown that Scots pine seedlings grown under RL exhibit greater photochemical activity of PSII and an increase in plant biomass compared to those grown under BL, WL or white fluorescent light [2]. One would expect that the addition of RL to sunlight would also cause similar changes. However, according to our data, neither additional RL nor RL+FRL led to increased PSII activity, which was maximal only under WL-supplemented conditions (Figure 5). In our experiments, the plants were grown under a fairly high light intensity, especially when supplementary light was used. Apparently, at such a high light intensity, photosaturation occurs, and for this reason, the difference among the compared spectral variants was not as pronounced. Moreover, the intensity and fraction of the spectrum of BL are important for the saturation of photosynthesis, as shown by the data of Hogewoning et al., 2010 [39]. The proportion and intensity of BL were the highest with the addition of WL (Figure 1), which was consistent with the high PSII activity observed in the Y(II) data of the samples treated with additional WL. In contrast to those in RL, photosynthesis and photochemical processes are known to be inhibited at very high BL intensities [40,41]. Therefore, under conditions of high light falling on plants, the photochemical activity of the plants under light with a high fraction of RL could be lower than that under light with a lower fraction of RL. However, when analyzing the processes of photomorphogenesis and growth, which depend not only on photosynthetic activity but also on many other factors, a different trend was discovered. In particular, many growth parameters were greater in the variants with additional RL or RL+FRL than in those with other lighting conditions (Figure 2 and Figure 3). In our experiments, high-intensity RL light was used, which caused a decrease in photochemical activity, as judged by the reduced Y(II) parameter. This apparently leads to the activation of a compensatory mechanism that consists of an increase in stomatal conductance. This regulation is reflected in increased stomatal conductance when plants are exposed to RL (Figure 5). In addition, the levels of all the photosynthetic pigments decreased in the current-year needles in the RL treatment (Table 2). The proposed mechanism for this phenomenon may be a change in the efficiency of water use by plants. Thus, RL, by reducing the efficiency of photosynthesis, likely by reducing the pigment content and photochemical activity of photosystem II (PSII), can stimulate an increase in stomatal conductance and, accordingly, increase the stomatal conductance. This indicates that, at a high RL fraction, the decrease in photochemical activity is compensated for through the mechanism of increased stomatal conductance, which contributes to the maintenance of photosynthesis. This observation contrasts with the results obtained previously in experiments with narrowband light, where simultaneous decreases in photosynthesis, transpiration and photosynthetic pigment content were recorded under RL exposure in Scots pine plants [2]. These data highlight the possibility of fundamental differences in exposure to narrowband RL and RL, in addition to sunlight, which has a fairly high proportion of BL.

Light quality also affects endogenous hormones, including those of coniferous plants [42]. Yang et al. (2015) [42] demonstrated the important role of gibberellins (GAs) in the response of pine plants to narrowband RL and BL. Based on data obtained from studies of the expression of hormone biosynthesis genes under the influence of RL and FRL in Scots pine, it is assumed that, in addition to GA, auxins play an important role [1,42]. We hypothesized that other hormones may also be involved in plant responses to RL+FRL. RL+FRL, but not RL alone, activated the expression of the biosynthesis and signaling genes of brassinosteroids (BRs) *BR-α-RED* and *BRZ2,* as well as the key transcription factors of the auxin signaling gene *ARF16* (Figure 6). Exogenous treatment with brassinosteroids can activate xylem formation in some pine species [43]. It is important to note that a similar response to RL+FRL in the current-year shoots was observed in the cambium, and in the roots, the combination of RL+FRL had the greatest effect on the transcription of the most studied genes (Figure 6B). In addition, we observed an increase in the expression of the negative component of the jasmonate signaling gene *JAZa*, which is consistent with our data on seedlings, as well as with work on the exogenous treatment of pine plants with jasmonic acid [44].

In the roots under RL and RL+FRL, the intensity of the expression of the key cytokinin signaling gene *HPT1* increased (Figure 6C). It can be assumed that, in this case, cytokinin signaling can be activated in the roots, which leads to the stimulation of the growth of the root system; this is supported by our data on the fresh weight of the young roots in the plants with additional RL (Figure 2). It can be assumed that the stimulation of endogenous brassinosteroids or the activation of the signaling of these hormones due to plant irradiation (RL+FRL) can have an effect similar to that of exogenous BR. Fan et al., 2021, showed that exogenous BRs affected xylem development more than phloem development. A similar statement is true for our data, where RL stimulated xylem development. It is highly likely that BR interacts with other phytohormones to regulate xylem development, as the levels of IAA, IAA, GA3 and BR were significantly altered in the stems of the BR-treated seedlings compared to those in the stems of the control plants [43].

Light quality also affects the formation of secondary metabolites, including in conifers [42]. The activation of the biosynthesis of several secondary metabolites by light under different spectral compositions has been observed in higher plants [22]. Most often, this occurs because photoreceptors can bind to the promoter regions of the genes for light-dependent transcription factors, such as *HY5* and *PIFs*, which, in turn, can change the expression of genes encoding key enzymes involved in the biosynthesis of phenylpropanoids, flavonoids and terpenoids [45]. During our experiments, an increase in the content of catechins and proanthocyanidins occurred exclusively in the current-year needles of the plants additionally irradiated with RL or RL+FRL compared to the control without the addition of WL or FRL (Table 1 and Table 2). An increase in the expression of terpenoid biosynthesis genes was also noted, but this change did not lead to a significant increase in the levels of these secondary metabolites (Figure 6 and Table 1 and Table 2). Apparently, the quality of light during the short-term cultivation of pine plants does not significantly affect the synthesis of terpenoids. It is worth noting that the contents of the majority of the studied phenolic compounds, flavonoids, catechins and proanthocyanidins tended to increase under RL in the current-year needles and roots but not in the cambium or previous-year needles (Table 1 and Table 2). It can be assumed that, in the case of a longer experiment, we could have observed an increased content of these compounds under these conditions (increased level and proportion of RL+FRL).

The significant role of the root system in supporting the vital functions of pinus seedlings must be acknowledged. However, the key growth parameters in the roots were notably greater (by 20–25%) in the RL treatment than in the control and WL treatment (Figure 2). We hypothesize that extending the experiment to a full year might reveal more significant differences. Additionally, with the addition of RL, an increase in the expression of hormone signaling genes in the roots and needles was observed. Although hormone levels in various tissues were not directly measured, the mutual increase in hormone signaling gene expression and cellular-level responses strongly suggests a potential alteration in the content of active hormones. This may also contribute to the elevated growth of plants under RL. Additionally, we detected the levels of TEAC, as well as the contents of catechins and proanthocyanidins, in the roots and current-year needles under RL, which were found to be 20% greater than those in the control and WL treatment groups. This increase could contribute to the enhanced resistance and, likely, the growth of the current-year roots.

## 5. Conclusions

Additional lighting with RL and FRL, used either individually or in combination, significantly affects the physiological, morphological and molecular characteristics of two-year-old Scots pine plants under conditions of supplemented sunlight. This effect is manifested in the increased development of xylem cells, changes in stomatal conductance and in the expression of genes that provide hormonal signaling via auxins and brassinosteroids in young (current-year) needles, as well as auxins and cytokinins in roots, against the background of jasmonate signaling inhibition.

It is worth emphasizing that such additional lighting also stimulates the synthesis of certain secondary metabolites, such as catechins and proanthocyanidins, thereby possibly improving plant resistance to various environmental stress factors. The use of a combination of RL and FRL under controlled conditions can serve as an effective tool for optimizing the growth of Scots pine in closed systems, which include greenhouses. We suggest that additional lighting (mainly FRL and RL+FRL) activates endogenous hormonal signaling in plants and, hence, the contents of key phytohormones, and such treatment with additional light under certain conditions is more effective than exogenous treatment with phytohormones.

## Figures and Tables

**Figure 1 cells-13-00194-f001:**
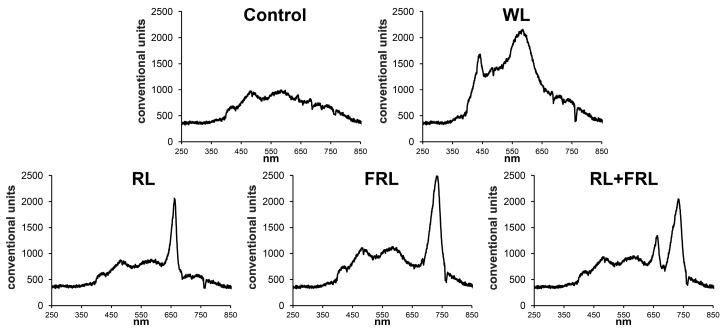
Light condition spectra used in the experiments.

**Figure 2 cells-13-00194-f002:**
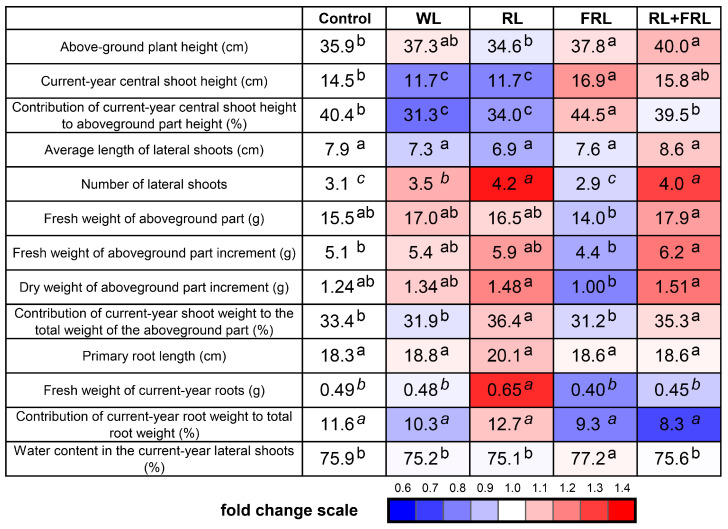
Morphometric measurements of plant growth under different light conditions. Different letters within each row indicate significant differences (*p* < 0.05) according to ANOVA followed by Duncan’s multiple range post hoc test (regular letters) or Kruskal–Wallis one-way ANOVA on ranks followed by the Student–Newman–Keuls post hoc test (italic letters). The values presented in the tables and figures are the arithmetic means ± standard errors.

**Figure 3 cells-13-00194-f003:**
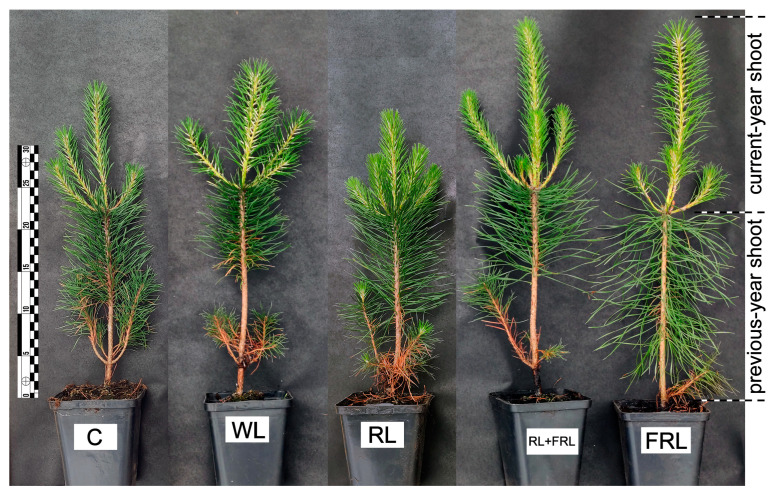
Appearance of the Scots pine plants at the end of the experiment.

**Figure 4 cells-13-00194-f004:**
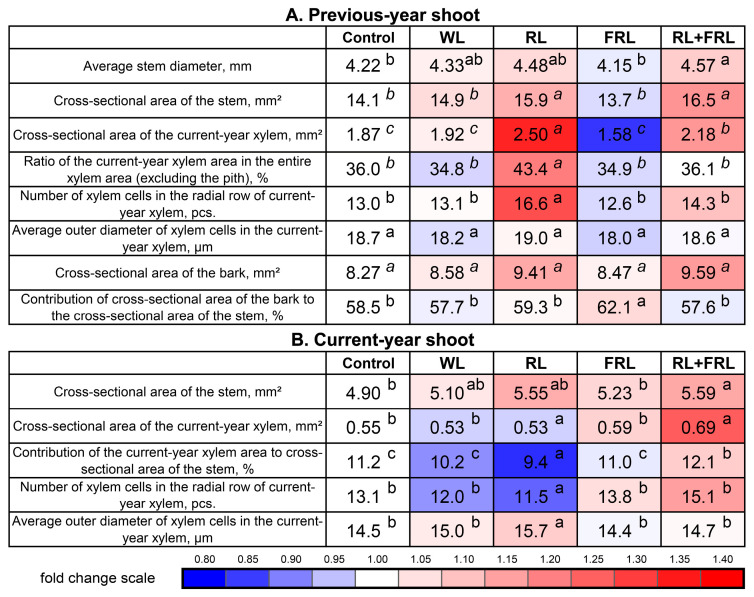
Anatomical parameters of the previous-year shoot (**A**) and current-year shoot (**B**). Different letters within each row indicate significant differences (*p* < 0.05) according to ANOVA followed by Duncan’s multiple range post hoc test (regular letters) or Kruskal-Wallis one-way ANOVA on ranks followed by the Student–Newman–Keuls post hoc test (italic letters). The values presented in the tables and figures are the arithmetic means ± standard errors.

**Figure 5 cells-13-00194-f005:**
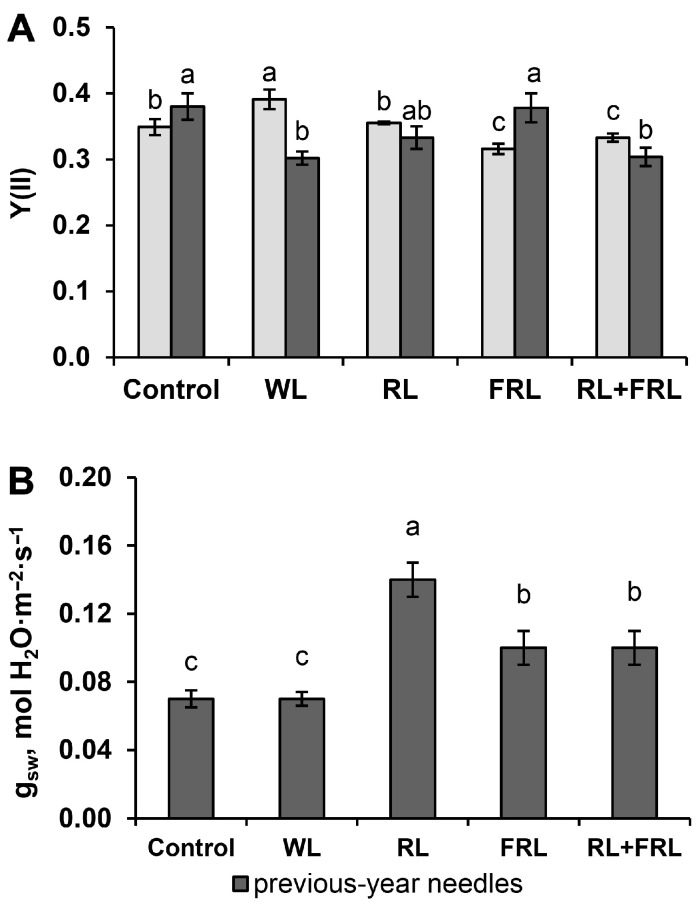
Effect of additional light on Y(II) (PSII effective quantum yield) (**A**) and stomatal conductance (g_sw_, mol H_2_O m^−2^s^−1^) (**B**) in previous-year and current-year needles of Scots pine. Different letters indicate significant differences (*p* < 0.05) according to ANOVA followed by Duncan’s multiple range post hoc test (regular letters). The values presented in the tables and figures are the arithmetic means ± standard errors.

**Figure 6 cells-13-00194-f006:**
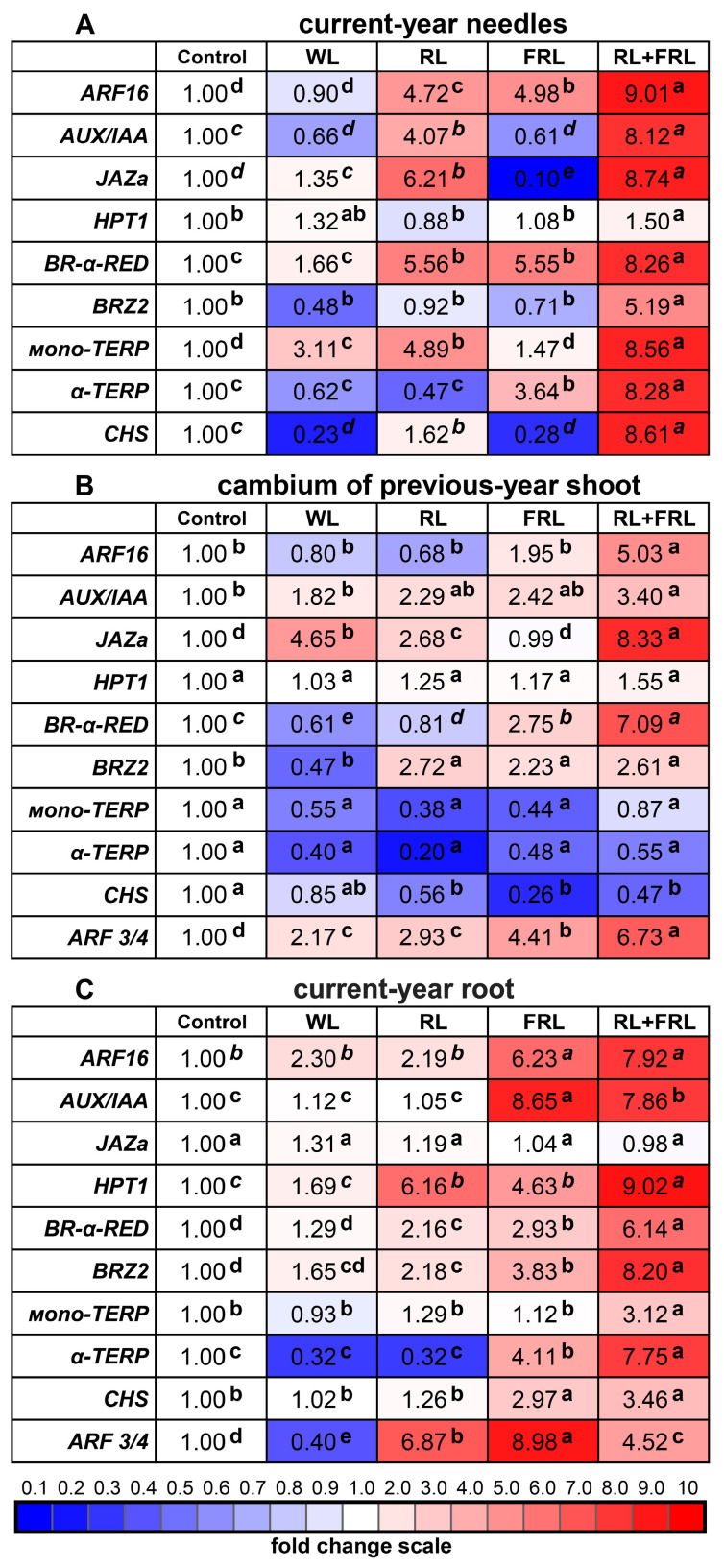
Effects of additional light on the expression of several hormone signaling genes, namely, auxin response factor 16 (*ARF16*), auxin response factor 3/4 (*ARF3/4*), auxin-induced protein 1 (*AUX/IAA*), jasmonate–Zim domain 1 (*JAZa*), histidine-containing phosphotransfer 1 (*HPT1*), Steroid 5-α-reductase (*BR–α–RED*) and brassinosteroid-mediated signaling pathway gene (*BRZ2*), and on genes involved in secondary metabolite biosynthesis, namely, monoterpene synthases such as TPS–mono1 (*mono-TERP*), α-terpineol synthase TPS–α-terp (*α-TERP*) and chalcone synthase (*CHS*) in current-year needles (**A**); cambium of previous-year shoots (**B**); and current-year roots (**C**). The color scale indicates how many times the expression changes: dark red represents a change of more than 2 times, and dark blue represents a change of more than 2 times relative to the control of sunlight without additional light. The transcript levels were normalized to the expression of the *Actin1* gene in the aboveground organs and to the expression of the *GAPDH* gene in the roots. Gene expression under sunlight without additional light was used as one unit. Different letters within each row indicate significant differences (*p* < 0.05) according to ANOVA on ranks followed by Duncan’s multiple range post hoc test (regular letters) or Kruskal-Wallis one-way ANOVA on ranks followed by Student-Newman-Keuls post hoc test (italic letters).

**Table 1 cells-13-00194-t001:** Effects of light of different spectral compositions on antioxidant capacity, phenolic compounds and total terpenoids in current-year roots.

Current-Year Roots	Control	WL	RL	FRL	RL+FRL
TEAC, µmol Trolox/g FW	98.8 ± 16.2 b	91.7 ± 11.0 b	113.7 ± 13.0 a	90.5 ± 14.4 b	101.5 ± 8.7 b
GAE, mg/g FW	5.05 ± 0.62 a	4.97 ± 0.55 a	5.85 ± 1.00 a	4.82 ± 0.63 a	5.31 ± 0.35 a
Flavonoids, mg catechin/g FW	5.25 ± 0.65 a	5.29 ± 0.49 a	6.07 ± 1.04 a	5.09 ± 0.78 a	5.48 ± 0.44 a
Catechins + proanthocyanidins, mg catechin/g FW	8.44 ± 1.33 b	8.36 ± 1.16 b	10.83 ± 1.23 a	8.41 ± 1.42 b	9.16 ± 0.76 b
Proanthocyanidins, mg cyaniding/g FW	2.02 ± 0.29 a	2.25 ± 0.27 a	2.64 ± 0.50 a	2.18 ± 0.34 a	2.38 ± 0.20 a

Different letters within each row indicate significant differences (*p* < 0.05) according to ANOVA on ranks followed by Duncan’s multiple range post hoc test. TEAC, Trolox equivalent antioxidant capacity; GAE, gallic acid equivalent. The values presented in the tables and figures are the arithmetic means ± standard errors.

**Table 2 cells-13-00194-t002:** Effect of light of different spectral compositions on antioxidant capacity, phenolic compound, total terpenoids and main photosynthetic pigment content in current-year needles.

Current-Year Needles	Control	WL	RL	FRL	RL+FRL
TEAC, µmol Trolox/g FW	62.8 ± 10.9 b	59.7 ± 4.5 b	79.8 ± 7.3 a	40.9 ± 2.6 b	65.0 ± 7.2 b
GAE, mg/g FW	3.85 ± 0.82 *a*	2.97 ± 0.30 *a*	4.26 ± 0.93 *a*	2.52 ± 0.41 *a*	3.23 ± 0.27 *a*
Flavonoids, mg catechin/g FW	0.68 ± 0.08 *a*	0.71 ± 0.08 *a*	0.94 ± 0.20 *a*	0.54 ± 0.08 *a*	0.75 ± 0.09 *a*
Catechins + proanthocyanidins, mg catechin/g FW	0.83 ± 0.15 *b*	2.06 ± 0.42 *a*	2.91 ± 0.48 *a*	0.70 ± 0.21 *b*	2.44 ± 0.62 *a*
Proanthocyanidins, mg cyaniding/g FW	0.37 ± 0.14 *b*	0.96 ± 0.16 *a*	1.15 ± 0.18 *a*	0.31 ± 0.13 *b*	1.20 ± 0.28 *a*
Terpenoids, mg/g DW	6.98 ± 0.23 a	6.98 ± 0.08 a	6.24 ± 0.31 a	6.74 ± 0.31 a	6.62 ± 0.20 a
Chl *a*, mg/g DW	2.50 ± 0.06 a	2.77 ± 0.17 a	1.85 ± 0.17 b	2.46 ± 0.05 a	2.54 ± 0.21 a
Chl *b*, mg/g DW	1.14 ± 0.03 a	1.23 ± 0.08 a	0.81 ± 0.08 b	1.23 ± 0.02 a	1.19 ± 0.10 a
Carotenoids, mg/g DW	0.53 ± 0.01 a	0.60 ± 0.03 a	0.43 ± 0.03 b	0.56 ± 0.01 a	0.55 ± 0.04 a
Chl *a*/Chl *b*	2.20 ± 0.02 b	2.27 ± 0.02 ab	2.32 ± 0.04 a	2.02 ± 0.03 c	2.13 ± 0.01 b
Carotenoid/chlorophyll *a* + *b* ratio	0.147 ± 0.002 *b*	0.151 ± 0.002 *b*	0.164 ± 0.004 *a*	0.151 ± 0.002 *b*	0.148 ± 0.002 *b*

Different letters within each row indicate significant differences (*p* < 0.05) according to ANOVA on ranks followed by Duncan’s multiple range post hoc test (regular letters) or Kruskal-Wallis one-way ANOVA on ranks followed by Student-Newman-Keuls post hoc test (italic letters). TEAC, Trolox equivalent antioxidant capacity; GAE, gallic acid equivalents.

## Data Availability

The datasets generated and/or analyzed during the current study are available from the corresponding author upon reasonable request.

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
