# Peer review of "Influence of Additional White, Red and Far-Red Light on Growth, Secondary Metabolites and Expression of Hormone Signaling Genes in Scots Pine under Sunlight"

_cells, 2024, doi:10.3390/cells13020194_

Round 1
Reviewer 1 Report
Comments and Suggestions for Authors
The manuscript reported functions of additional lighting with RL and FRL or combining of RL+FRL on morphological and physiological characteristics. (note: reviewer does not sure whether the conditions of FRL+RL is the same as RL+FRL).
The methods section gave solid details for experimental design and data analysis. Some data presented in tables could be converted into graphic presentation, much easy to read, especially, for comparing among treatments. At least, authors should consider highlighting the highest value of each parameter in table 1. same for table 2.
There are some concerns about discussion and conclusive statements.
Line 461-462, 'FRL light stimulate the elongation of the central shoot', was supported well by the data in Table 1, however, no information about 'increased water content' as one reason for this elongation. according to the data in Table 1, FRL showed the lower water content (fresh weight - dry weight) compared with that from RL and RL+FRL treatments.
Line 465, mentioned 'water content', again (see above), authors cited Table 1.
in the conclusion, authors suggested that additional lighting conditions of FRL and FRL+RL, however, the data presented in current manuscript did not include FRL+RL (not RL+FRL). Genral knowledge supports that the ratio of active/inactive phytochrome depends on the later lighting treatments.
Author Response
Dear Reviewer,
Thank you for your valuable comments and insights. We appreciate the opportunity to clarify and improve our manuscript based on your feedback.
- The manuscript reported functions of additional lighting with RL and FRL or combining of RL+FRL on morphological and physiological characteristics. (note: reviewer does not sure whether the conditions of FRL+RL is the same as RL+FRL).
Answer: We would like to express our appreciation to the reviewer for pointing out that the phrases 'FRL+RL' and 'RL+FRL' are equivalent. This observation helped us identify a typographical error in our text. To ensure clarity and avoid any potential confusion among readers, we have amended this mistake accordingly.
- The methods section gave solid details for experimental design and data analysis. Some data presented in tables could be converted into graphic presentation, much easy to read, especially, for comparing among treatments. At least, authors should consider highlighting the highest value of each parameter in table 1. same for table 2.
Answer: We sincerely appreciate the reviewer's constructive feedback regarding the presentation of the data in Tables 1 and 2. We acknowledge the suggestion to enhance the readability and comparative clarity of our findings by highlighting the highest value of each parameter within these tables. It is done.
- Line 461-462, 'FRL light stimulate the elongation of the central shoot', was supported well by the data in Table 1, however, no information about 'increased water content' as one reason for this elongation. according to the data in Table 1, FRL showed the lower water content (fresh weight - dry weight) compared with that from RL and RL+FRL treatments.
Line 465, mentioned 'water content', again (see above), authors cited Table 1
Answer: We extend our gratitude to the reviewer for identifying this ambiguity in our manuscript. Acknowledging the lack of clarity, we have revised the text to ensure that it conveys our intended message more accurately and unambiguously.
We improved the text as follows: “Our studies also indicate that FRL light stimulates the elongation of central shoots (current-year central shoot height; Table 1), probably due to cell elongation and an increase in the percentage of water content of 77.2 ± 0.3 (Figure 2; Table 1). The increase in growth through extension under the influence of supplemental FRL was substantiated by the observed increase in metrics such as aboveground plant height, current-year central shoot height, and the contribution of the current-year central shoot height to the aboveground part height under these light conditions. This difference was additionally confirmed by the relative decrease in fresh biomass in the RLB treatment compared to that in the RL treatment (Figure 2; Table 1). »
- In the conclusion, authors suggested that additional lighting conditions of FRL and FRL+RL, however, the data presented in current manuscript did not include FRL+RL (not RL+FRL). Genral knowledge supports that the ratio of active/inactive phytochrome depends on the later lighting treatments.
Answer:
We appreciate the Reviewers’ attention to this detail. Importantly, our study focused on supplementary irradiation, where we added extra far-red and red light to natural sunlight, as detailed in the Materials and Methods section. We acknowledge the reviewer's reference to the concept of RL-FRL reversibility, wherein the sequence of light exposure is crucial. While conducting such experiments in the absence of sunlight might indeed offer a more straightforward approach to studying reversibility, our study did not aim to explore this aspect. Our objective was to examine the effects of additional light wavelengths in conjunction with natural sunlight.
Reviewer 2 Report
Comments and Suggestions for Authors
The manuscript entitled “Influence of Additional White, Red, and Far-Red Light on Growth, Secondary Metabolites, and the Expression of Hormone Signaling Genes in Scots Pine under Sunlight” is well written and the important information are reported on relevant topics, although I suggest changing the title, because the plants were grown under diffused sunlight in greenhouse: “Influence of Additional White, Red, and Far-Red Light on Growth, Secondary Metabolites, and the Expression of Hormone Signaling Genes in Scots Pine under Diffused Sunlight”
The experimental design is simple, but it was clearly and correctly conducted, although the tissue sampled for gene analyses expression appear not clear, due to different age of sample tissues analyzed. The data processing is partially correct, in fact, the analysis run on the percentage data should be transformed in arcsin or arctan before the ANOVA analysis. The description and discussion of results are completed and clearly argued: the figure 1 missed some information on the relative value (relative to what?).
Many other criticism and suggestion are directly reported below.
I do not recommend the publication of the manuscript in the present form, so I suggest to publish the manuscript after the correction and the revision of the manuscript, and after the provided information which has been requested.
The manuscript suffers for the poor description of Material and Methods section.
Line 50 delete “.. as well as RL/FRL. ..”
Line 51 correct in “…fraction of phytochrome molecules .. “
Line 108 Correct in pinene and limonene,
Line 162, 164, 165, 167 and 168, Correct in: μmol m–2 s–1…..
Line 224, correct in: Trolox equivalent antioxidant capacity (TEAC) ..
Line 312-314: WHAT ABOUT THE VALUE OF THE TOTAL ROOT WEIGHT? Percentage data should be converted in arcsin or arctan before the ANOVA analysis! Indicate which value represent the number after the average: standard error? Standard deviation?
Line 341, 352 Authors followed by Duncan’s method. Percentage data should be converted in arcsin or arctan before the ANOVA analysis!
Line 415: Authors, what about the Stomatal conductance (gsw)?
Line 417 “Gene expression was studied in ..” The authors' choice of tissues and organs for the expression analysis is not clear. Why have you chosen different organs with different age? Do the authors have any motivation for this choose?
Line 419 Correct in: RL and joint “action of (RL+FRL) of light on ..”
Line from 421 to 435, Caption of Figure 4. Why have you used post hoc test only in some of the gene expression analysis? Change the title of Figure 4C from young root to current-year roots.
Line 442 “The level of gene transcripts in cambium cells was similar;” It is not true! The expression pathways of genes is not similar.
Line 453-455. Regards the following sentence “.. It should also be noted that the level of expression of the auxin signaling gene ARF3/4 was highest in the FRL variant, in contrast to other studied organs and tissues (Figure 4C).” You should consider that expression analyses of this gene only concern current-year roots. The expression level under FRL and RL+FRL conditions is higher in the tissues studied. Correct this statement.
Considering the statement you make on lines 491-494 “ An increase in the number of xylem cells in the current-year shoot under the influence of RL+FRL indicates activation of cell division, while RL reduces the number of cells in the current-year shoot and simultaneously increases their diameter (Table 3) .” It would have been interesting to test the expression of HTP1 in the bud change of the year, to assess the cytokinin pathway.
Line 595: The plants were not “under two-year-old Scots pine plants under conditions of supplemented sunlight”, because you have used shade net to diffuse the direct sunlight. The shade net changed the quality of light and the relative relationships between different wavelengths.
Author Response
Dear Reviewer,
Thank you for your valuable comments and insights. We appreciate the opportunity to clarify and improve our manuscript based on your feedback.
- The manuscript titled “Influence of Additional White, Red, and Far-Red Light on Growth, Secondary Metabolites, and the Expression of Hormone Signaling Genes in Scots Pine under Sunlight” is well written, and important information is reported on relevant topics, although I suggest changing the title because the plants were grown under diffuse sunlight in a greenhouse: “Influence of Additional White, Red, and Far-Red Light on Growth, Secondary Metabolites, and the Expression of Hormone Signaling Genes in Scots Pine under Diffused Sunlight”
Answer: We express our sincere gratitude to the reviewer for their suggestion and agree with their perspective. However, we would like to note that the impact of this suggestion is somewhat unclear, given that the experimental setup has already been comprehensively described in the Materials and Methods section of our manuscript.
- The description and discussion of results are completed and clearly argued: Figure 1 missed some information on the relative value (relative to what? ).
Answer: Thank you for the note, not relative units but conventional units.
- The manuscript suffers for the poor description of Material and Methods section.
Line 50 delete “…as well as RL/FRL...”
Line 51 correct in “…fraction of phytochrome molecules .. “
Line 108 Correct in pinene and limonene,
Line 162, 164, 165, 167 and 168, Correct in: μmol m–2 s–1…..
Line 224, correct in: Trolox equivalent antioxidant capacity (TEAC) ..
Line 419 Correct in: RL and joint “action of (RL+FRL) of light on ..”
Answer: This was done.
- Line 312-314: WHAT ABOUT THE VALUE OF THE TOTAL ROOT WEIGHT? Indicate which value represent the number after the average: standard error? Standard deviation?
Answer: We improved the Statistical section 2.8 and the Table captions as follows: “The values presented in the tables and figures are the arithmetic means ± standard errors.
- Line 341, 352 Authors followed by Duncan’s method. Percentage data should be converted in arcsin or arctan before the ANOVA!
Answer: We appreciate the Reviewer for your comment regarding the statistical treatment of the percentage data in our manuscript, specifically in lines 341 and 352. While we recognize the debate surrounding the transformation of percentage data for ANOVA, our approach was guided by the capabilities of SigmaPlot software. This software automatically determines the appropriateness of applying ANOVA or ANOVA on ranks, depending on the adherence to normal distribution requirements. We have explicitly mentioned this procedural detail in the Materials and Methods section, as well as in the captions of relevant tables and figures. Therefore, we believe our current methodological approach is justified and consistent with the software’s analytical framework.
- Line 415: Authors, what about the Stomatal conductance (gsw)?
Answer: We improved the information throughout the text.
- Line 417 “Gene expression was studied in ..” The authors' choice of tissues and organs for the expression analysis is not clear. Why have you chosen different organs with different age? Do the authors have any motivation for this choose?
Answer: We appreciate the opportunity to clarify our methodological choices regarding tissue selection for gene expression analysis. Our initial intent was to exclusively analyse young needles formed under the varied light conditions of our experiments. However, as the study progressed, we became interested in exploring how gene expression might differ in other plant organs. Regarding the stem tissue, we observed that a significant portion of the previous year's stem comprised dead xylem cells, rendering it impractical for a comprehensive gene study. Consequently, we chose to analyse the cambium from the last year's stem, as this was the most viable option. Separating the cambium of the current year is technically challenging and requires microscopic intervention, which was beyond the scope of our study. We decided against including needles from the previous year in our analysis. On the basis of our extensive experience with conifers under greenhouse conditions, we found that these needles predominantly reflect the light conditions present during their formation. This historical signal complicates the interpretation of the current experimental results. Similarly, old roots were excluded from our analysis. Separating these roots from their growing substrate is a complex process, and like last year's needles, they tend to carry information from previous years rather than reflecting the current experimental conditions. This historical influence could confound our analysis of the current light impact.
- Line from 421 to 435, Caption of Figure 4. Why have you used post hoc test only in some of the gene expression analysis? The title of Figure 4C was changed from young roots to current-year roots.
Answer: We carried out this test everywhere. We have made adjustments to the captions: instead of Duncan's method, we wrote Duncan's multiple range post hoc test and in the Materials and Methods section.
- Line 442 “The level of gene transcripts in cambium cells was similar;” It is not true! The expression pathways of the genes were not similar.
Answer: This was done.
- Line 453-455. We regard the following sentence “. It should also be noted that the level of expression of the auxin signaling gene ARF3/4 was highest in the FRL variant, in contrast to that in other studied organs and tissues (Figure 4C). You should consider that expression analyses of this gene only concern current-year roots. The expression levels under FRL and RL+FRL conditions were greater in the tissues studied. This statement has been corrected.
Answer: This was done.
Improved text: “It should also be noted that, in contrast to those in other studied organs and tissues, the level of expression of the auxin signaling gene ARF3/4 in current-year roots was highest in the FRL and RL+FRL variants (Figure 4C).
- Considering the statement you make on lines 491-494 “ An increase in the number of xylem cells in the current-year shoot under the influence of RL+FRL indicates activation of cell division, while RL reduces the number of cells in the current-year shoot and simultaneously increases their diameter (Table 3) .” It would have been interesting to test the expression of HTP1 in the bud change of the year, to assess the cytokinin pathway.
Answer: We express our gratitude to the reviewers for their insightful recommendations. Our study initially intended to focus on the buds from the current year. Specifically, we investigated dormant buds from the previous year that revived during spring under varying spectral light conditions. Our primary analysis encompassed the cellular structure of stems sprouted from these buds in the current year. Additionally, we examined young needles at an early developmental stage prior to the commencement of our experiment.
While a detailed study of the differentiating apex tissue is feasible and an exploration of the apex would indeed be intriguing, these aspects extend slightly beyond the scope of the present study. However, we intend to incorporate these elements in our future research endeavors
- Line 595: The plants were not “under two-year-old Scots pine plants under conditions of supplemented sunlight”, because you have used shade net to diffuse the direct sunlight. The shade net changed the quality of light and the relative relationships between different wavelengths.
Answer:
Thank you for your observation. We endeavoured to ensure that the grid minimally altered the spectral composition and intensity ratio of the light. The grid was constructed from white, nonwoven fabric, specifically, from the form of flilin, which reflects the entire light spectrum without modifying its specific composition. Furthermore, as all the plants were uniformly covered, this introduces a systematic error that can be mitigated. Notably, greenhouse glass likely filters out a certain percentage of UV light, which is generally undesirable. However, as these conditions were consistently applied across all the experimental procedures, we have duly noted them in the Materials and Methods section of our study.
Reviewer 3 Report
Comments and Suggestions for Authors
The manuscript titled: “Influence of Additional White, Red, and Far-Red Light on Growth, Secondary Metabolites, and the Expression of Hormone Signaling Genes in Scots Pine under Sunlight” by Pashkovskiy and coworkers, submitted to Cells indicated that additional red (RL) and far-red (FRL) light can accelerate the development of pine plants by increasing the number of xylem cells and by the increase in aboveground parts, but not by rising the photosynthetic activity nor by the accumulation of secondary metabolites. According to the authors the additional RL +FRL increased the expression of several genes responsible for the biosynthesis and signaling of auxins and brassinosteroids, while the expression of the signaling pathway gene of the jasmonic acid was reduced. It is well known that hormones and secondary metabolites play an important role in increasing plant resistance especially by protecting plants against reactive oxygen species. The effect of plant response to changes in the RL/FRL ratio was also well described in the literature. High level of RL/FRL induces auxin synthesis and polar auxin transport, promoting shoot elongation while cytokinins stimulate cell division and differentiation, especially in roots, thereby promoting a balanced growth. Changes in the RL/FRL ratio can also affect the formation of diverse classes of secondary metabolites such as terpenoids, which also play an important role in plants' response to stress. The authors formulate a hypothesis that the effect of this combination of various spectra of light acting through auxin transport causes the increase in the number of xylem cells. To prove the hypothesis, different anatomical measurements were performed for the current-year as well as for the previous-year shoots such as: stem diameter, cross section area of xylem cells, the number of xylem cells in the radial row etc. Transpiration and fluorescence parameters as well as content of phenolic compounds, analysis of total terpenoids, and RNA extraction and RT‒PCR analysis were also performed. All data were statistically correctly analyzed. Morphometric measurements of plant growth under different light conditions indicated that RL+FRL caused a substantial increase of height of current-year shoots (as shown on Figure 2). However, the anatomical measurement showed different effects of RL and FRL light combination on various parameters measured in the experiment. Also phenolic, terpenoids, photosynthetic pigment contents did not differ significantly in different light spectral composition. The effect of additional light on the expression of some genes of different hormones was also not clear. Most of the differences obtained were not really significant. The results obtained by the authors clearly indicated that combination of additional light treatment leads to different effects at the structural, physiological and molecular levels. This is not astonishing because these effects are strongly connected to each other. An increase of one parameter causes an decrease of another parameter. I would like the authors to convince me that such analysis is of great value and that: “The use of a combination of RL and FRL under controlled conditions can serve as an effective tool for optimizing the growth of Scots pine in closed systems, which include greenhouses”. Concluding: In my opinion the data included in the manuscript: “Influence of Additional White, Red, and Far-Red Light on Growth, Secondary Metabolites, and the Expression of Hormone Signaling Genes in Scots Pine under Sunlight” by Pashkovskiy and coworkers are not convincing and therefore I don’t recommend the paper to be published in Cells.
Author Response
We express our gratitude for the reviewer's thorough examination of our manuscript and acknowledge the concerns raised. The effect of the red-to-far-red light ratio on herbaceous plants is well documented, while its impact on gymnospermous woody plants is a burgeoning area of research. Since 2012, our team has actively contributed to this field, consistently monitoring the global literature on the subject. Our previous studies conducted under hydroponic culture conditions within a climate chamber using narrow-band red light indicated a 100% increase in xylem cell count. It is recognized that far-red light can alter auxin levels, promoting elongated growth. In the manuscript, we explore the possibility that combining red and far-red light exposure could simultaneously increase cell count and size, with our findings suggesting that this is achievable, albeit with certain caveats. We also investigated the amplification of plant secondary metabolism and flavonoid and terpenoid formation under supplemental illumination. It appears that light intensity, more than spectrum, significantly influences most parameters, as evidenced by the lack of statistical variance from other treatments in terms of the effects of white light. Notably, red light and a combination of red and far-red light had discernible impacts on molecular and cellular aspects, albeit not on photosynthesis or secondary metabolite synthesis, as detailed in our manuscript. We attribute this to the duration and environmental variability of the experiment. Our study, though revealing noticeable effects, may have been too brief for a woody gymnosperm plant. An extension to late autumn might uncover how pine plants adapt to changing growth periods, potentially influencing gene expression and secondary metabolite formation after a certain period. Our laboratory's resources were also considered. The comprehensive processes of plant growth, section preparation, and biochemical and molecular analyses were labor intensive and resource demanding. Nonetheless, we assert that red and combined red and far-red light have a positive influence on pine plants under greenhouse conditions. While this may not be immediately apparent from the data, we continue to advocate for this hypothesis.
This text was added to the Discussion section:
Additionally, with the addition of RL, an increase in the expression of hormone signaling genes in roots and needles was observed. Although hormone levels in various tissues were not directly measured, the mutual increase in hormone signaling gene expression and cellular-level responses strongly suggest a potential alteration in the content of active hormones. This may also contribute to the elevated growth of plants under RL. Additionally, we found that the levels of TEAC, as well as the contents of catechins and proanthocyanidins in the roots and needles in recent years under RL, were 20% greater than those under the control and WL treatment. This increase could contribute to the enhanced resistance and, likely, the growth of the current-year roots.
In response to the reviewer comments, we transferred Table 5 and part of Figure 3 to the supplementary section and clarified certain data points in Tables 1, 2, and 3. As a result, we detected significant differences among the variants, especially in the root system (by 20–25%). This difference occurred during growth for 6 weeks.
We hope these amendments satisfactorily address the reviewer’s concerns.
Round 2
Reviewer 2 Report
Comments and Suggestions for Authors
Now it is ready for publication
Reviewer 3 Report
Comments and Suggestions for Authors
I thank the authors for all changes and corrections made in the manuscript. I appreciate the mitigating and softening interpretation of differences in light spectrum and their combinations. Therefore, the differences emphasized in the first version are more reliable in the corrected version.
Taking the above into account I conclude that the manuscript : “Influence of Additional White, Red, and Far-Red Light on Growth, Secondary Metabolites, and the Expression of Hormone Signaling Genes in Scots Pine under Sunlight” by Pashkovskiy and coworkers, has been sufficiently improved to warrant publication in Cells.